# Perspectives of Motor Functional Upper Extremity Recovery with the Use of Immersive Virtual Reality in Stroke Patients

**DOI:** 10.3390/s23020712

**Published:** 2023-01-08

**Authors:** Paweł Sip, Marta Kozłowska, Dariusz Czysz, Przemysław Daroszewski, Przemysław Lisiński

**Affiliations:** 1Department of Rehabilitation and Physiotherapy, Poznan University of Medical Sciences, 28 Czerwca 1956 Str., No 135/147, 60-545 Poznań, Poland; 2Wiktor Dega Orthopaedic and Rehabilitation Clinical Hospital, 28 Czerwca 1956 Str., No 135/147, 60-545 Poznań, Poland; 3SciTech, Zbąszyńska Str., No 7/7, 60-359 Poznań, Poland; 4Department of Organization and Management in Healthcare, Poznan University of Medical Sciences, Przybyszewskiego Str., No 39, 60-356 Poznań, Poland

**Keywords:** stroke, virtual reality, upper limb, motor function, mirror therapy

## Abstract

Stroke is one of the leading causes of disability, including loss of hand manipulative skills. It constitutes a major limitation in independence and the ability to perform everyday tasks. Among the numerous accessible physiotherapeutic methods, it is becoming more common to apply Virtual Reality "VR”. The aim of this study was to establish whether immersive VR was worth considering as a form of physical therapy and the advisability of applying it in restoring post-stroke hand function impairment. A proprietary application Virtual Mirror Hand 1.0 was used in the research and its effectiveness in therapy was compared to classical mirror therapy. A total of 20 survivors after ischaemic stroke with comparable functional status were divided into a study group (n = 10) and control group (n = 10). Diagnostic tools included 36-Item Short Form Survey “SF-36” and the Fugl-Meyer Assessment Upper Extremity “FMA-UE”. Collected metrics showed a normal distribution and the differences in mean values were tested by the student’s *t*-test. In both, the study and control groups’ changes were recorded. A statistically significant outcome for FMA-UE and SF-36 measured by the student’s *t*-test for dependent or independent samples (*p* > 0.05) were obtained in both groups. Importantly, proven by conducted studies, an advantage of VR proprietary application was subjective sensations amelioration in pain and sensory impressions. Applying Virtual Mirror Hand 1.0 treatment to patients after a stroke appears to be a good solution and definitely provides the opportunity to consider VR applications as an integral part of the neurorehabilitation process. These results give a basis to plan further larger-scale observation attempts. Moreover, the development of the Virtual Mirror Hand 1.0 as an innovative application in physiotherapy may become equivalent to classical mirror therapy in improving the quality and effectiveness of the treatment used for post-stroke patients.

## 1. Introduction

Nowadays, it is inevitable to apply digital solutions in clinical health care. The intensification of this process is provided by the worldwide tendency to develop technological solutions. This transfers into the possibility of gathering data, validating the performance of a patient’s tasks and accurate reflexion of the demanded movement. Based on the clinical experience, the continuation of the rehabilitation process is often practiced at home, which could be successfully resolved by Virtual Reality “VR” appliances and the possibility to remotely monitor the patient’s condition [1,2,3,4]. A significant part of numerous functional stroke symptomatology is manipulative skills dysfunction [5]. This impairment appears as a stroke consequence in almost 60% of cases and can last for more than 12 months [6]. Many years of clinical observations allow for the establishment of a thesis that the restoration of manipulative function is a complex and demanding task [7]. Therefore, various descriptions dedicated to this issue can be found in the literature which provide methods that specify their effectiveness. In the last decade, the use of a botulinum toxin followed by exercises [8], vibration training [9,10], kinesiotaping [11], electrostimulation [12], use of dynamic splint [13] and constraint-induced movement therapy were postulated in rehabilitation [14,15]. The results of research dedicated to mirror therapy being used in order to deal with manipulative skills impairments appear promising [16,17,18,19]. Recently, due to computer technology development, successful use of VR in improving fine motor skills in stroke patients is observed [20,21,22,23]. The mentioned scientific reports are describing the effects of a particular method in relation to the control group, in which it is excluded from the rehabilitation program. However, we have not found any research that would compare the above-mentioned methods giving evidence of the advantage of one over the other in the literature. We assumed that in order to improve the manipulative hand functions, it is beneficial to combine a traditional approach with innovative computer technology, known as virtual rehabilitation or VR [24]. The treatment basis of classic mirror and VR application therapies focuses on the same principle: the patient sees his impaired hand moving and an exact imitation of the not-plegic side motion is presented either in the mirror reflection or in the head-mounted display. VR is an image of “artificial reality” created by multiple devices interconnected by a computer system. Therefore, as a classic approach was already a theme of multiple research projects, we decided to compare the effects of mirror therapy and VR application. A VR system uses the role of visual feedback to make stimuli reaching the patients as realistic as possible. Mei-Hong Zhu et al. devoted their attention to this issue [25]. In the traditional approach to the subject, VR application in the rehabilitation field not only needs dedicated computer software, but also devices that display and collect information about the patients’ movement. The display equipment includes traditional computer monitors, LCD screens and projectors [26]. The most modern systems, Cave Automatic Virtual Environment “CAVE”, represent a high-tech solution, in which projectors present a stereoscopic image on the walls and floor of the room. Patients using this system need to wear stereoscopic glasses to be able to view 3D images [27]. With a second kind of display device being glasses or Head-Mounted Display “HMD”. In order to conduct therapy more effectively, equipment that detects a patient’s movement and provides biofeedback in the form of an image is required. This is possible due to a motion detector or 3D cameras giving the patient the possibility to react and to fulfil a task appearing on the screen or a console [28]. Depending on the display equipment used, there exist VR types: VR with immersion (immersive VR), augmented VR and Mixed Reality “MR”.

In the first one, the activation of proprioceptive sensations let the patient feel that he is being transferred to another multisensory environment, that helps patient keep his attention on a given task. This effect can be obtained due to a HMD-type display placed in a helmet or glasses that isolate the person from their surroundings. When a sound or an avatar character that reflects the patient’s movement is added, an even greater immersion effect can be achieved [29]. We decided to choose that VR type according to the above-mentioned reasons. 

In the second type of VR, augmented VR, the user sees both the natural environment and virtual characters or objects placed. This technology integrates application content into real-world settings [30].

In the third type of environment, MR, the latest sensory solutions and imaging technologies are used. The patients register both images displayed by LCD screens or projectors and other objects and people present in the room; thus, receiving stimuli from the virtual world and natural environment [4]. 

As a conclusion, most beneficial solution for the post-stroke patient might be immersive VR. That VR type provides isolation from the real environment as an essential feature in focusing attention and the correct interpretation of the movement. What is more for the patient, there are no distractive factors, but only a pure image regardless of external stimuli. Adherence to neurotherapy and optimising therapeutic aims can be effectively achieved [4]. Due to dedicated software, the therapy scheme as well as recorded results are saved in the memory of VR device. This allows for the observation of the progress of treatment, to capture weak the point of therapy and to plan the next stages of rehabilitation. VR application use is a modern technology solution simplifying the therapeutic process and collecting a great database in order to conduct diagnostics and to be used in further research [29].

The aim of this study was to establish whether immersive VR was worth considering as a form of physical therapy and the advisability of applying it in restoring post-stroke hand function impairment. Our clinical concept is derived from widely known in the neurorehabilitation mirror therapy.

## 2. Materials and Methods

### 2.1. Participants

Randomized studies on 20 patients of Neurological Rehabilitation Department “STROKE” in Wiktor Dega Orthopaedic and Rehabilitation Clinical Hospital in Poznan were conducted from July 2022 to October 2022. The following inclusion criteria were defined and enforced during the qualification assessment:-diagnosis of first-episode stroke,-age range 40–64,-acquired motor impairment of hemiplegic upper limb,-maximum of 12 months period since diagnosis,-functional brain damage specified with Rankin scale 1–4 at the last hospital discharge.

Certain exclusion criteria were admitted:-requirement of constant, intensive medical surveillance,-active comorbidities significantly influencing rehabilitation process (ex. bone fractures occurred during medical treatment, pressure ulcers, etc.),-circulatory insufficiency, kidney, liver failure, condition after myocardial infarction with ejection fraction less than 30%,-vascular disease (active thromboembolism),-heart aneurysm, aortic aneurysm, malformation of cerebral vessels,-active inflammation,-uncompensated endocrine disruption,-cancer (palliative care or need of urgent treatment),-severe arterial or pulmonary hypertension,-uncontrolled diabetes,-epilepsy.

The above-mentioned criteria were made in order not to disturb or to be a risk during the neurorehabilitation process.

The study was conducted according to the guidelines of the Declaration of Helsinki and approved by the Ethics Committee of Poznan University of Medical Sciences (protocol code 587/22, date of approval 23 June 2022). Informed written consent was obtained from all subjects involved in the study. Written informed consent has been obtained from the patients to publish this paper.

In this research, we selected patients from the “STROKE” Department and then randomly divided into two, equal in terms of number study and control group. 

Twenty patients with hemiplegia dexter according to computer tomography scan and 3.1 ± 0.57 points in the study (min-max, 2–4) and 3.3 ± 0.67 points in the control group in Rankin Scale (min-max, 2–4) were included in the trial. Mean time after right-sided stroke diagnose (and occurrence) caused by right medial cerebral artery ischaemia resulting in plegic left upper limb was 3.4 ± 1.43 months in the study and 3.3 ± 0.67 points in the control group. All participants of this study were after the first stroke and took part in the first rehabilitation program. The mean age of patients in the experimental group was 54.9 ± 3.98 years, in the control group it was 59.2 ± 4.34 years, and in both groups, it was 57.05 ± 4.62 years. Duration of the research for each group, consistent with the “STROKE” project establishments, was 18 days and occurred in three consecutive weeks, form Monday to Saturday. Each participant of the research was assessed with quality-of-life scale 36-Item Short Form Survey “SF-36” and related to sensorimotor function of upper limb Fugl-Meyer Assessment Upper Extremity “FMA-UE” before the therapy starts. 

### 2.2. VR Application 

Patients in the study group followed a physical therapy treatment of upper limb with the use of SciMed system which includes the immersive VR application Virtual Mirror Hand 1.0, implemented on the Oculus Quest 2 VR glasses module (Figure 1).

Oculus Quest 2 is a VR system designed and manufactured by Meta, former Facebook, the most popular VR set in 2021, does not require any computer connected to work and is one of the most affordable VR sets on the market. Furthermore, the device can track hands position and gestures without any additional hardware. The application uses Oculus Quest 2 unique sensors to track a patient’s healthy hand movement and visualize the paretic hand which is not in motion. In this technology, before the training program starts, the physiotherapist indicates non-paretic hand. It is a pattern for the sensors to copy on a paretic hand. Hand position is represented by joint positions and rotations. The SciMed application calculates the theoretical position of the patients’ paretic hand, as it is a mirror reflection of a healthy hand. User sees both hands moving equally.

During exercising (Figure 2), patients wearing Oculus Quest 2 were set into unreal, VR imitating outer space (Figure 3, Figure 4 and Figure 5). There is no need to customize the training, as the built-in cameras automatically recognize patients’ hands and adjust an image basing on that information. Application was generating a view of both hands, ready to start the movement. Logging in the mobile device, the practitioner could register and constantly monitor performed sequence of patients’ movements checking if the therapeutic exercises are proceeded correctly. 

Control group was undergoing a classic mirror therapy treatment (Figure 6 and Figure 7) that uses a reflection seen in the mirror to proceed upper limb treatment. 

### 2.3. Research Procedures

Before the research procedure start, patients, besides objective assessment test (FMA-UE and SF-36), were asked for subjective sensations in paretic hand (pain, finger tingling, fingertips numbness, heat sensation, fingertip sensation, impression of movement and movement). In both, VR and mirror therapy group, the not-impaired hand was a reference point and was determining normative values in the therapeutic procedure of the opposite side. Mentioned value was referring to the subjective movement pattern including velocity, trajectory and range of motion determined by the not-plegic side. 

Before the experimental study, each patient was separately taught the correct manner, number of series and repetitions of exercises performed by the not-impaired hand. 

The protocol for both, study and control group, lasted for 18 days. It was performed for 30 min and included 20-s breaks between the series of exercises. Patients were asked to do exercises presented on Figure 1.

During the therapy, exercises were repeated by the practitioner when required. They were proceeded in and were performed in fixed order. Subjects were provided with the same environmental conditions as for the place and external stimuli. After finishing of 18 days rehabilitation program, each patient form both study and control group were assessed with FMA-UE and SF-36. 

### 2.4. Statistical Analysis 

The statistical analysis was performed with open access online programme and MS Excel 2016. The descriptive statistics were reported as mean value with standard deviation, median, min-max. In both experimental and control group participating in the research, scores and mean values obtained in FMA-UE and SF-36 were analysed in two-time points, before the therapy and after 18 days of treatment. All collected data were checked for normal distribution with the Shapiro–Wilk test. Provided normal distribution, data was compared with the paired *t*-test. The paired *t*-test was conducted to compare the differences between the results obtained before and after exercise period within the groups and then between the groups. *p*-values of less than 0.05 were considered statistically significant.

## 3. Results

Collected metrics showed normal distribution and the differences in mean values were tested by paired *t*-test. Mean difference in points between M1 and M2 measurements in FMA-UE concerning study group was 7.5 ± 4.7 and was statistically significant. Statistical significance regarding the same scale was also confirmed by the result of 6.0 ± 4.4 between M1 and M2 in control group. The SF-36 questionnaire score, in the study group, reached 19.1 ± 21.4 and was statistically significant. Corresponding result from control group was 5.3 ± 6.1 and there was statistical significance relation noted. 

To complete data gathered from above mentioned tests, there was also taken into consideration mean pain value of upper limb registered during passive movement performed by a practitioner. Diminution of feeling pain occurred in both groups; however, a statistically significant improvement was noticed only in the experimental group and was 1.6 ± 2.5. Exact values and measurement results are presented in Table 1.

In addition, in study group, while performing exercises, 9 out of 10 patients reported one of the following symptoms: fingers tingling, fingertips numbness, heat sensation, impression of fingertips sensation and of real movement in the hand. During therapy, three patients began to perform a slight fingers flexion that was perceptible and recorded by physiotherapist. This movement occurred when forming a fist with non-paretic side. The above-mentioned action was confirmed by a practitioner as a flicker of movement that was also interpreted in VR application as a change in positioning of healthy hand. Subjective symptoms were not present before the treatment started. That can suggest a great patient’s involvement towards the treatment. In control group, any of the symptoms listed in Table 2 were reported by only two patients. 

Table 3 presents the patient’s answers to the question about the knowledge of the current form of therapy. In the control group, none of the patients have been treated with VR technology previously. In the control group, 8 out of 10 people reported having heard of mirror therapy treatment.

## 4. Discussion

VR has increasingly become an integral part of medicine and motor dysfunction therapy [24]. Therefore, we also decided to attempt development in this direction by implementing a new, author’s Virtual Mirror Hand 1.0 application for upper limb on Oculus Quest 2.0 to treat sequelae of stroke. Clinically proven by ours and Israely et al.’s and Mutai et al.’s research, stroke patients presenting fine motor functions disability, have major limitations in everyday, professional and social life [7,31]. Depending on impairment grade, nervous system regenerative abilities, the intensity of therapy and patient’s involvement in treatment, we can evaluate whether the given rehabilitation process is successful. It is worth regarding how to intensify the therapeutic procedure to achieve the most satisfactory effect. It is worth considering that the musculoskeletal consequences of stroke are most effectively treatable in up to three months following stroke. In addition, collateral to a motor disability, we observed multiple cognitive skills impairments, negative impact on the emotional sphere resulting in focusing attention difficulty, frequent lack of hope towards recovery as well as weakness in body perception. The above-mentioned changes were also observed by Grefkes et al. and Wade et al. [32,33]. Therefore, we asked ourselves how much of the proposed and implemented immersive VR therapy for stroke patients can prove valuable in these considerations to effectively influence on the above clinical signs. Additionally, most importantly, whether it can ultimately improve the quality of treatment in this group of patients.

Rankine scale is commonly used in determining and interpreting the degree of disability, including stroke patients. This scale was used in our study as well as in the observations of Zupanic et al. [34].

We gathered and interpreted data from objective patient condition questionnaires in the analysis. FMA was used to evaluate upper limb motor function and SF-36 questionnaire was applied to measure psychophysical health. The same measurement tools were implemented and considered as valuable in many other clinical trials presented by Zondervan et al. and Zhao et al. [35,36]. All subjective information and impressions provided to the specialist by patients in the study and control groups were also recorded.

It may be noted that none of the subjects in the study group had prior contact with the VR being a form of a treatment. The lack of popularity of such advanced virtual treatments can be confirmed by the fact that there have been very few scientific reports published over the last 5 years [37]. Otherwise, mirror therapy was known to 80% of neurological patients in the control group; additionally, many references can be found in the literature. Kil-Byung et al. and Wen et al. evaluate patients after stroke with upper limb motion deficits using mirror therapy as one of the treatment methods [38,39]. It may also be confirmed by research that mirror therapy is suggested not only to be more easily available to patients, but also possible to perform independently at home [40].

After completing a therapeutic program, we noticed changes in patient’s FMA-UE and SF-36 results. FMA-UE improvement occurred in both, the study and the control group; therefore, VR and classic mirror therapy result in a positive upper limb functionality change. Similar effects were presented by Weber et al. and Thieme et al. [29,41]. In our study, all evaluated patients presented a better SF-36 score. We can note a resemblance of that outcome in the studies conducted by Aramaki et al. [42]. All our data shows a statistical significance, thereby indicating the amelioration of functioning and quality of life. Another important factor influencing stroke patients’ general condition is an upper limb pain level measured before and after the treatment. Mean difference for those metrics were statistically significant only in the study group. VR therapy efficacy in pain reduction is proven by Spiegel et al. [43]. According to our knowledge, treatment performed in VR environment is worth considering for further observations in terms of pain tolerance. As an effect, it could contribute to wider upper limb movement capabilities, which is obviously an issue for stroke patients. Mobility limitation in upper limb joints caused by feeling pain that was evaluated by us in our group of patients is also confirmed by Nickel et al. [44]. Despite objective and revealing numerous similarities results in both groups, subjective sensations cannot be omitted. They were very explicitly described and underlined by tested patients and were concerning sensory impressions on plegic hand. They have been noticed during the therapy and at the end of the treatment day. One of the accompanying subjective symptoms was the movement of plegic limb confirmed by a specialist performing and observing the therapeutic procedure. In most cases, those sensations were concerning the experimental group. Although the concept of classical mirror therapy and VR treatment remains the same, the environment, where therapy is performed may have great influence on its effects. 

From a technical point of view, we can assume that there are many factors influencing the effectiveness of the therapeutic effects. In the study group, this could be isolation from the surrounding environment, no distracting aspects, focus on a specific task, realistic image of affected upper limb hand in the application, ease of instructing and understanding the exercises by immediate biofeedback response, the attractiveness of therapy as modern technology. As immersive VR is a brand-new method, observations constitute a ‘novum’ and requires further research. However, subjective, observable and reported symptoms in the plegic hands of patients during and immediately after VR treatment may suggest targeted, appropriate nervous system stimulation. Our above-mentioned assumptions may be validated by the research conclusions conducted by Sungbae et al. [45]. 

We are aware of the VR appliance limitations in therapy. Only 20 patients complied with the strict inclusion and exclusion criteria of our study. Longer treatment observations would be beneficial. The application itself also requires enhancements and upgrades with more accessible diagnostic capabilities such as recording collected data on the patient profile, introducing patient-specific motion tasks and many more. However, our satisfactory results of Virtual Mirror Hand 1.0 therapy give us the basis for further observation and development of the application itself with new therapeutic features and measurement capabilities. Although, given that much more changes in data values have been observed in the group of patients with our own treatment than in the case of traditional mirror therapy, we would like to be very hopeful that perhaps we are in the hands of a new, post-stroke treatment tool for patients. We are also clearly aware of the huge development opportunities ahead of us, and these can also be directed toward delivering the treatment in all circumstances, mainly at home, with full, remote monitoring of the patient in real time by a specialist.

## 5. Conclusions

Applying Virtual Mirror Hand 1.0 treatment to patients after stroke appears to be at least as good as classical mirror therapy solution and definitely provides the opportunity to consider the VR application as an integral part of neurorehabilitation after stroke. The VR and classical mirror therapy beneficial properties are evaluated in objective tests and questionnaires. However, only VR therapy patients reported improvement in pain and multiple subjective sensations. The VR application is intuitive, accessible to understand and easy to learn from the very beginning. It was well perceived and tolerated by all respondents, even though it has been the first contact with VR for all of them. The development of the Virtual Mirror Hand 1.0 system is likely to be equivalent to increasing the quality of the treatment and delivering even greater benefits for patients.

## Data Availability

The data presented in this study are available on request from the corresponding author.

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
