# Peer review of "Perspectives of Motor Functional Upper Extremity Recovery with the Use of Immersive Virtual Reality in Stroke Patients"

_sensors, 2023, doi:10.3390/s23020712_

Round 1

Reviewer 1 Report (Previous Reviewer 1)

Thank you for the interesting work. The authors presented the work that studies the effect of novel immersive VR-based motor neurorehabilitation for stroke patients.

The work is structured well and it is written well. However, I have some remarks:

Introduction

> Depending on the display equipment used, there exist VR types: VR with immersion (immersive VR), augmented VR and VR without immersion (non-immersive VR). In the first one, the activation of proprioceptive sensations let the patient feel that he is being transferred to another multisensory environment. This effect can be obtained due to HMD-type display placed in a helmet or glasses that isolate the person from surroundings. When a sound or an avatar character that reflects the patient’s movement is added, an even greater immersion effect can be achieved.

I invite authors to specify what kind of VR they have used in the study and why.

> Due to dedicated software, the therapy scheme as well as recorded results are saved in the memory of VR device. This allows to observe the progress of treatment, to capture weak point of therapy and to plan next stages of rehabilitation. VR application use is a modern technology solution simplifying the therapeutic process and collecting great database in order to conduct diagnostics and to be used in further research [25]. Mul- tiple advantages of VR system and its possible technological superiority over classic mirror therapy can be found in chapter 2.2.

I was expecting to see the aim of the paper at the end of the Introduction. I invite authors to clearly state the aim of the work.

Applying Mirror Hand 1.0 treatment to patients after stroke appears to be at least as

good as classical mirror therapy solution and definitely provides the opportunity to con-

sider the virtual reality application as an integral part of neurorehabilitation after stroke.

The VR and classical mirror therapy beneficial properties are evaluated in objective tests

and questionnaires. However, only VR therapy patients reported improvement in pain

and multiple subjective sensations. The VR application is intuitive, accessible to under-

stand and easy to learn from the very beginning. It was well perceived and tolerated by

all respondents, even though it has been their first contact with virtual reality for all of them.

The development of the Mirror Hand 1.0 system is likely to be equivalent to increasing

the quality of the treatment and delivering even greater benefits for patients.

Another suggestion regards the name of the application. By reading the conclusion, I found confusing the name of the VR application "Mirror Hand 1.0" and I would associate it more with standard therapy (the name) rather than VR. I suggest some more clever names, such as VR Mirror Hand or virtual mirror hand, or something similar.

I invite authors to address some issues in the discussion. For example, the 20 subjects is a considerable sample, but usually not enough (at least statistically) to make certain claims. I see that at the beginning both groups have a similar ranking scale  "3.1 ± 0.57 vs 3.3 ± 0.67", however, I see that the inclusion criteria are 1-4. I mean, that is a broad range which starts from no significant disability despite symptoms to moderately severe disability; unable to walk and attend to bodily needs without assistance. As the group of subjects is small, and the mean might not be always representative (except the scores are not Normally distributed), I suggest either putting max and min values or even better reporting a table with a ranking scale for all subjects as supplementary material.

I would also like to suggest the authors to authors take a look at the motor-imagery-based control + VR (10.1371/journal.pone.0250431 10.1016/j.procs.2020.09.270 10.1088/1741-2552/aba162/meta), which might be implemented in a similar manner.

Author Response

Reviewer 2 Report (New Reviewer)

Motor functional upper extremity recovery with the use of immersive virtual reality in stroke patients: a comparative clinical trial

In this article virtual reality works no better than mirror therapy for the upper extremity   in patients with stroke. Comparisons were conducted within group, but no comparisons were conducted between groups. The analyses of the results as well as their discussion should be improved considering the comparisons between groups.

Lines 48-52 : for a common criterion and for comparisons, is it possible to know the improvement obtained by these different interventions?

Lines 53-54 : "appear promising" : cf. previous comment. What improvement is expected comparative to the other interventions?

Lines 100-101 : motion sickness is observed in older persons. Maybe explain here some disadvantages associated with VR.

Lines 109: What is the aim of this study? Could you specify it?

Lines 113: considering 10 patients by group, what is the power of this study?

Line 120: with which level of impairment?

Lines 150-157: Results section?

Lines 157-159: bring this info close to those in lines 143-146?

Lines 160-164: Results sectuib?

Lines 165-167: maybe collect all outcomes in a specific sessions (evaluation) describing them (main outcome, secondary outcomes)

Line 170: has this system been used in previous studies? If so, complete the introduction with those results (?).

Lines 177-178: maybe inform the price?

Line 188: VR

Lines 209-211: cf. comment about description of assessments and outcomes

Lines 240-243: analysis.

Lines 243-250: for both outcomes, the main results should be the comparison after treatment for interventional and control group

or

comparison between differences M1-M2 between groups.

Lines 251-253: cf. comment about "evaluation section"

Table 1: complete the results with comparisons between groups?

If results are the same for both groups, what is the relevancy of RV against classical mirror therapy?

Lines 266-282: please, previous inform these evaluations in Methods (?).

Lines 289: the acronym "VR" was used previously and should be used until the end of the paper.

Please, reviwe the use of the acronyms (?).

Discussion should be modified according with the comments above and focused about the main results of the study.

Author Response

Reviewer 3 Report (New Reviewer)

Round 2

Reviewer 1 Report (Previous Reviewer 1)

The authors have addressed all my concerns and therefore I support the publication of the article.

Reviewer 2 Report (New Reviewer)

No further comments

Reviewer 3 Report (New Reviewer)

No other comments

This manuscript is a resubmission of an earlier submission. The following is a list of the peer review reports and author responses from that submission.

Round 1

Reviewer 1 Report

Dear Authors, 

Thank you for the interesting work. The authors presented the work that studies the effect of novel immersive VR-based motor neurorehabilitation for stroke patients.

The work is structured well and it is written well. However, I have some remarks:

Introduction

1. Depending on display equipment used, there exist two VR types: VR with immersion (immersive VR) and VR without immersion (non-immersive VR). In the first one, the patient feels that he is being transferred to another reality, he is immersed in it.

I invite the authors to further elaborate on the immersive part. The term immersive is explained as "immersed in it", which does not provide more information. I suggest improving it in more technical terms (e.g. proprioceptive). 

2. Material and methods

I invite the authors to give more information on the type of stroke.

maximum of 12 months period since diagnosis 

Does the diagnosis correspond to the time of the stroke occurrence? 

functional brain damage specified with Rankin scale 1-4 at the last hospital discharge. 

1-4 is very broad, can you provide more information on the mean+-SD of the scores for the experimental vs control group? 

Results

SF-36 were analysed twice 

I would suggest changing it to two-time points. 

slight fingers flexion that was perceptible and recorded in reality

In virtual reality or was observed by the physiotherapists?

4 out of 5 people reported previous experience with mirror therapy. 

Does that mean that they have a previous history of stroke, or did they start with rehabilitation before the study started?

Overall nice and very interesting work. I suggest 2nd round of the review to mainly clarify the methodological aspects of the study population.

I would also like to suggest the authors to authors take a look at the motor-imagery-based control + VR (10.1371/journal.pone.0250431 10.1016/j.procs.2020.09.270 10.1088/1741-2552/aba162/meta), which might be implemented in a similar manner.

Author Response

Good morning,

Thank you for such a quick review. We appreciate your precious remarks and we applied them immediately to our manuscript.

Introduction. We rewrote the sentence explaining the immersive Virtual Reality in more technological way. (line 83-85)

Material and methods. We understand and confirm, that we did not specify the group accurately enough. We added some details regarding participants choice to conduct the studies. “STROKE” Department had much wider including criteria, but for this research patients were carefully selected and then randomly divided into two groups. Both study and control group were homogeneous regarding impaired side, time of stroke occurrence as well as diagnosis, type of stroke and disability level. Number of participants has changed since the beginning (from 18 to 14) due to personal factors that did not allow the patients to stay at the ward. (line 131-138, 140-143)

Results:

-Thank you, we changed the expression “twice” for “to two-time points”. (line 224)

-“Reality” means “by physiotherapist”. We clarified that in the manuscript, as it may actually be confusing to use the word “reality”. Thank you for that remark. (line 254-256)

-We clarified that previous experience with mirror therapy means a knowledge of that kind of post-stroke treatment appliance. We changed that information both in table 3 and in the results section (line 264-265, 269).

Thank you for suggesting us an adequate to our work article, that will definitely be useful for our further observations and studies.

Your review meaningly enriched our outcome and gave us an opportunity to be more precise about the material and methods.

Reviewer 2 Report

This study reported that the mirror hand application using immersive VR improved motor function in chronic stroke patients. However, it could not reveal that mirror therapy using VR was superior to conventional mirror therapy. There are major problems with the small number of participants, the lack of adjusting of age in the two groups, and the lack of direct comparison of the effects themselves.

Methods

You should analyze the difference in age between the two groups. It seems that the control groups were older than the VR groups. Isn’t it natural that young patients with stroke are more likely to recover than old patients?

The number of control groups is too few to analyze the effect of conventional mirror therapy. Isn't it natural for 5 participants not to show a significant difference?

Please describe the VR application Mirror Hand 1.0 in more detail. Did you customize it?

Please describe the instruction of paretic hands when patients moved the non-paretic hands.

Discussion

Scientific research conducted by Soto-Cámara et al. confirms that the risk of stroke is clearly increasing beyond 55 years and affects men slightly more than women [30]. Similar data were also observed in our study.

This explanation should be deleted because the aim of your study isn’t connected with it.

Author Response

Good morning,

Thank you for your review. We analyzed your ideas to improve our manuscript. We found you remarks extremely useful. We relate to each of your suggestions.

Study and control group

We understand and confirm, that we did not specify the group accurately enough. We added more details regarding participants choice to conduct the studies. “STROKE” Department had much wider including criteria, but for this research patients were carefully selected and then randomly divided into two groups. Both study and control group were homogeneous regarding impaired side, time of stroke occurrence as well as diagnosis, type of stroke and disability level. Number of participants has changed since the beginning (from 18 to 14) due to personal factors that did not allow the patients to stay at the ward. Therefore, only 14 participants followed our study program without any disturbances. (140-143)

What’s also important, our studies describe an innovative technology appliance, so the main aim of our work at that time was to have preliminary report on this kind of post-stroke treatment, what may explain the small number of participants.

“Lack of adjusting of age” à Thank you for your attention put on the age factor. It is an important value in our study. According to our knowledge and experience also provided by following research https://pubmed.ncbi.nlm.nih.gov/15118964/, our attention was not so much focused on the age diversity in our groups, as it remains uncertain to be a factor in recovery process. What’s more, after the 4 participants resignation, the mean age of control group has slightly increased. Obviously, we find your remark very valuable and we would put more emphasis on it in out further research to avoid doubts (line 140-143)

“Lack of direct comparison of the effects themselves” à We suggest it was an initial study to have an overall idea on possible results that can be achieved. Therefore, the results in the research group revealed greater changes in comparison control group. We suppose, such result gives us basis to perform further work in the field. (line 229-231)

Methods

-There is no need to customize the training, the VR Application does it itself using built-in cameras. (line 172-174)

 -We explained that by “forming a fist” instruction, that patients performed with his non-paretic side, patients started moving their paretic hand fingers. (line 255)

Discussion

The main aim of this study was not to describe the age of patients having stroke. You have a point, thank you for that suggestion, there is no direct observation in our studies regarding age of the patients. It appears useless to mention that in our study. (line 291-293 deleted)

Your review meaningly enriched our outcome and will be very useful in our further research giving us an idea what to focus and pay attention on. 

Reviewer 3 Report

This article is written in a medium to low English level, making it difficult to understand. 

The article is about using an immersive VR  application to perform mirror therapy on patients who have had a stroke. Its abstract gives us an idea of the quality of the article. For example, in line 20, the word "appliance" does not make sense in the sentence. Also, statistical significance is mentioned, but there is no description of the statistical test used or to which metric it was applied.

In virtual reality literature, non-immersive VR is known as augmented VR, so this should be made explicit in the text (line 80).

Was there an ethical committee approving the clinical trial? If there was, then this should be explicitly mentioned in the text. If there was no approval, the research lacks medical endorsement.

The paragraph that describes how mirror therapy is performed using the VR system is confusing (lines 146-161). First, it should be explained what mirror therapy consists of and then what metaphors were used for its implementation.

For statistical analysis, normality tests were applied to the metrics collected; in the case of normality, a  t-student test for mean differences is done (which is not said explicitly). It is also not explicitly stated whether all metrics passed the normality test or what would be done if a metric did not have a normal distribution.

The Results section found significant differences in the means of the FMA and SF-36 tests for the experimental group. Also, some symptoms were reported during the exercises in members of both groups. It is not understood what is meant by M1 in Table 2. 

Given the small number of members in the control and experimental groups, the statistical power of the t-student test should be determined to get an idea of the reliability of the estimate of the probability that the means of the measurements are significantly different.

Author Response

Good morning,

Thank you very much for your review and precious suggestions. We corrected our manuscript considering your advice. We explained doubts and misunderstandings appearing in our work.

We confirm our inaccuracy in abstract. Word “appliance” was replaced by “application” (line 20), “non-immersive” by “augmented” (line 81,89). We added detailed information about statistics as you reasonably noticed. (26-28)

The research obviously received the submission from Medical University of Poznan Ethics Committee with the number 587/22. We delivered the scan of this document to the editorial office. We added that information to the manuscript as well. (line 128-130)

Thank you for paying attention on the explanation of how the Mirror Hand 1.0 application works. We added a description which can directly be transferred on mirror therapy assumption. (line 165-170)

Collected metrics showed normal distribution and the differences in mean values were tested by t-student test. (line 25,204)

Before the research procedure start, patients, besides objective test, were asked for subjective sensations in paretic hand. That outcome is reported as M1 in table 2. (line 192-193)

Our studies describe an innovative technology appliance, so the main aim of our work at that time was to have preliminary report on this kind of post-stroke treatment, what may explain the small number of participants.

Your review meaningly enriched our outcome and will be very useful in our further research giving us an idea what to focus and pay attention on. We realise it was an initial study to have an overall idea on possible results that can be achieved. We suppose, such result gives us basis to perform further work in the field.

Round 2

Reviewer 1 Report

>Depending on display equipment used, there exist two VR types: VR with immersion (immersive VR) and VR without immersion (augmented VR). 

I don't believe that augmented reality is always without immersion. I think that it depends on the content and applications.

>All subjects presented post-ischemic stroke condition according to computer tomography scan and 3 or 4 points on the Rankin scale.

It is important to specify the mean ranking scale for each group, experimental and control, as it was performed with age. It is also advisable 

to include other clinical and demographic parameters to understand if the two groups are the same from the start.

>Mean time, after right-sided stroke diagnosis (and occurrence) caused by right medial cerebral artery ischaemia resulting in plegic left upper limb, was 3 months.

In connection to the previous question, it is advisable to include also the information for each patient simply because the mean might not be representative, especially when it is not written repeatable for the experimental and control groups.

> Both study and control group were homogeneous regarding impaired side, time of stroke occurrence as well as diagnosis, type of stroke and disability level.

I invited authors to specify the type of stroke, not because I dought the homogeneity of the group, but because of the replicability of the study, as I believe it is important scientific information for a reader

> We clarified that previous experience with mirror therapy means knowledge of that kind of post-stroke treatment appliance. We changed that information both in table 3 and in the results section (line 264-265, 269).

Does that mean that none of the patients had any standard motor rehabilitation program before VR treatment?

Reviewer 2 Report

Thank you for your answers to my questions and comments.

However, I am afraid that you did not show any direct comparison of the age between the two groups or direct comparison of the effects between the two groups. 

The number of control groups is too few to analyze the effect of conventional mirror therapy. As I repeatedly pointed out, it is natural for 5 participants not to show a significant difference.

Reviewer 3 Report

The comments made to the article were taken into account and have been corrected.

Therefore, the article is ready to be published.